# Lived experience of people on anti-retro viral therapy in the context of covid-19: A phenomenological study

**Tadele Derbew Kassie**[1]*, **Yosef Wasihun**[2], **Abiot Aschale**[1], **Fentie Ambaw**[2]

**1** Public Health Department, College of Medicine and Health Science, Debre Markos University, Debre Markos, Ethiopia, **2** Health Promotion and Behavioral Science Department, School of Public Health, College of Medicine and Health Science, Bahir Dar University, Bahir Dar, Ethiopia

* tadelederbew@gmail.com

## Abstract

### Introduction

People living with human immune virus (HIV) are confronting multiple psychosocial and economic issues influenced by the illness. People on anti-retro viral drugs (ART) were at risk for discontinuation of medications during corona viral disease-2019 (COVID-19) pandemic. COVID-19 outbreak made people living with HIV (PLWH) to experience critical challenges and barriers to optimal care. The experience of people living with HIV such as stigma and discrimination, economic problem, psychosocial problem before the emergency of COVID-19 were studied but there is lack of understanding on the lived experience of people living with HIV in the context of COVID-19 in Ethiopia particularly in Tach Gayint. This study aimed to explore the lived experience of people on ART in context of COVID-19 in Tach Gayint district.

### Methods and materials

Phenomenological study design conducted with 16 study participants from March 12-April 12/2021. Data were collected using in-depth interview using interview guide and digital recorder. The investigator took note in addition to digital record. Interviews were transcribed word for word and translated conceptually. Analysis followed Reading of transcriptions, develop and apply coding, displaying data, data reduction and interpretation. Atlas.ti-7 software used to facilitate analysis. The quality of data was assured by the principles of credibility, dependability, conformability and transferability.

### Result

The study explored psychological experience, change in social interaction, and economic experience and ART drug interruption as the main themes including other sub-themes. Most participants perceived the severity of COVID-19 on them. Lack of participation in social activities due to fear of contracting and the participants experienced dying and loss of income. This extreme fear pushed some participants to interrupt their daily ART intake

**Funding:** The author(s) received no specific funding for this work.

**Competing interests:** The authors have declared that no competing interests exist.

especially during lock down period. Personal, familial and community context contributed for these distressing experiences.

## Conclusion and recommendation

This study reported participants' psychosocial, economic experiences including ART drug interruptions. The government should design policies and interventions to alleviate their personal, household and community problems, which were the caused for the mentioned hostile experiences.

## Introduction

Anti-retroviral therapy (ART) involves taking a combination of HIV medicines called HIV treatment regimens every day and maximally suppressive ART regimens should be taken to obtain the best result [1]. People living with HIV/AIDs are confronting multiple psychological issues influenced by the illness and social attitudes towards them. As 38 million people living with HIV [2], patients of COVID-19-HIV co-infection are likely to increase. COVID-19 brought different challenges to people on highly active anti-retro viral therapy (HAART) [3]. People on ART were at risk for discontinuation of HAART and they did not know where to get ART drugs in the near future [4]. By the end of 2019 only 67 percent were accessed to ART drugs [2]. In Ethiopia it is estimated that the prevalence of HIV in 2017 was 1.16% and in the end of 2017 there were more than 41 hundred thousand adults on ART. And in 2020 the number of total people living with HIV is expected to be 745,719 [5].

Corona virus 2019 (VOVID-19) is a family of sever acute respiratory syndrome (SARS) caused by a novel respiratory virus that mostly affect the respiratory system of human and was first originated in Wuhan, China in December 2019 [6]. Globally the mortality rate of COVID-19 over time using a 14-day delay estimate was 5.7% [7]. Globally up to 15 January 2021, there were more than 91 million confirmed cases of COVID-19, including two million deaths, and in Africa there were more than two million confirmed cases reported to world health organization [8]. In Ethiopia the first COVID-19 case was detected in March 2020 and there were 120 hundred cases and 1,864 deaths reported until 23 December 2020 [9].

In March 2020, the Centers for Disease Control and Prevention (CDC) warned that people living with HIV (PLWH) as a population may be at heightened risk for severe physical health illness from COVID-19 compared to the general population [10]. The prevalence of COVID-19 on people living with HIV infection was 4.5% [11]. COVID-19 has potential interference with HIV care and treatments, and high rates of socially-produced burdens in the form of violence, stigma, discrimination, isolation, and hate on ART people [12]. People on ART obligated to uphold the lockdown and prevent transmission of COVD-19 which cause tension and anxiety on them; these combined effects resulted in delays accessing HIV care, missing clinic appointments and poor medication adherence. The rate of hospital admission of people HIV infection due to COVID-19 is higher compared to those who have no HIV infection [13]. In China, people on ART missed their HIV medication and there were high COVID-19 patient's overburdened disruption of medical care as the health system. To address this concern the government designed mailing ART to people living with HIV (PLWH) and this service made unwanted exposer of their status to their family [14]. In Europe there was no closer of ART clinic but they give only ART drugs and other service related to HIV care were not given and health professionals reported as there was shortage of ART medication and they may stop

ART drug procurement for the future as a result of COVID-19 [15]. During COVID-19 pandemic in Wisconsin, United State people on ART miss their medication more than one time per a week has 7% increment than that was before COVID-19 [16]. Physical distance that was taken as a measure of COVID -19 prevention mechanism affect the interaction between ART providers and ART users as a result ART adherence is reduced [17]. The current measures to control COVID-19 epidemic neglected the economic, social and health care service of people taking ART drugs [18]. In China, COVID-19 outbreak made people living with HIV (PLWH) to experience critical challenges and barriers to optimal care outcomes and due to the growing number of COVID-19 cases and the shortage of medical resources, [14]. People living with HIV expressed feelings of anxiety, fear, unhappiness and hopelessness to the future which are related to their illness during COVID-19 epidemic [19, 20]. In Belgium and Brazil during COVID-19-epidemic among people living with HIV 23.3% were positive for major depressive disorders, whereas 22.7% had generalized anxiety disorders [21]. During COVID-19, people on ART faced psychosocial problems like food insecurity due to lose of job opportunity severe depressive and anxiety symptoms in Kenya [22]. In Ethiopia youth people on ART assumed as there was no any service in health institutions and they feared to go clinic to bring ART medication as a result they stopped taking their ART medications [23]. There are researches done on the experience of people on ART such as stigma and discrimination, economic problem, psychosocial problem before the emergency of COVID-19. So far, the mortality rate and hospital admission rate of people on ART with COVID-19 studied. However, the lived experience of people on ART in context of COVID-19 pandemic not explored and there is lack of understanding on the lived experience of people on ATR in context of this pandemic in Ethiopia particularly in Tach Gayint district and qualitative study may be suitable to explore it. Currently, it is implemented to prevent the transmission of COVID-19. For better implementation of COVID-19 prevention mechanism on people living with HIV/AIDS, it was very critical to explore what they felt and what they experienced related to the emergency of COVD-19. Therefore this phenomenological study was aimed at an exploration potential experiences felt by people on ART in context of COVID-1 and understanding their feelings deeply in the district.

## Methods

### Study settings and period

This study was conducted in Tach Gayint district, South Gondar zone; Amhara regional state, Ethiopia from March 12-April 12/2021. There were six health centers, one primary hospital and two private clinics in the district. The government in the district was not doing any special activities, which supported people on ART in relating to COVID-19 to reduce the impact of COVID-19 on them. There were two ART trained health professionals, one ART adherence counselor and one data clerk working in ART outpatient department in the health center.

### Study design

Descriptive phenomenological approach was used. This technique explore the world through the word of a person. It seeks to investigate the meanings that the individual has encountered in everyday life and to arrive at a new view of the universe by uncovering new and overlooked meanings of these encounters. This approach gives deeper data from individuals' latent and unseen experiences, which are less accessible through other methods. In this study descriptive phenomenology design was used to explore the experience of people on ART in context of COVID-19 pandemic from March 12-April 12/2021.

## Participants

Study participants were those people who were on ART at the time of data collection and on ART before the emergency of COVID-19. Using heterogeneous purposive sampling, 16 participants who have appointment for ART, who come for different services in the health facilities during data collection period were recruited in the study. Their age ranged between 25 and 65 years. This sample size was determined based on data saturation. The reason why participants who were on ART before COVID-19 pandemic and on ART in data collection period selected was that they could tell their experience by comparing what their experience looked like before COVID-19 and at the time of COVID-19 pandemic. In addition, seriously ill participants who were unable to communicate and unable to give full information at the time of data collection were excluded. Participants were selected purposively through heterogeneous method to attain maximum variation of experience in context of COVID-19 in terms of different background characteristics.

## Sampling technique and participants requirement procedure

To select participants, meeting was made with care providers in the health center. At the meeting, the principal investigator explained the aim and objective of the study. Two health care providers were assigned to arrange and contact study participants with principal investigator. These care providers were the instrumental in facilitating the meeting between the principal investigator and study participants. The principal investigator explained the aim and objective of the study to the participants. He also informed them taking part in the study would mean they would be interviewed about their personal experience taking ART drug in context of COVID-19 and the interview would be audio recorded with their permission. With those who volunteered to participate, an agreement on convenient time and places were made for the interview.

## Data collection tools

In-depth Interview Guide and digital recorder were used to explore the lived experience of people on ART in context of COVID-19.

## The context of the study

After explaining the purpose, risks, benefits of the study and the length of the interview, the participants were asked to sign the consent in order to participate in the study. Principal investigator did sixteen In-depth Interviews. Fifteen in-depth interviews were held in consultation room provided by the health center based on their choice. The room provided by the health center was private, familiar and comfortable to the study participants. One in-depth interview was held in the participant's house. Two in-depth interviews per day were conducted. Face to face interview was done which allowed the researcher an opportunity to write memos. The interviewer requested permission to start the interview and digital recording to ensure verbatim transcription. Interviews were conducted until conceptual saturation (to the point no further new information was gained any more). Data collector used probing technique, by using how and why, to get adequate data on the point of interest. Participants were coded as P1, P2, P3. . ...respectively based on the order of the interview. Interviews were conducted using Amharic (local language). Sixteen in-depth interviews (IDI) were conducted and each IDI lasted an average of 65 minutes.

## Trustworthiness

To achieve the trustworthiness of research, study was done in line with the initial study design. Credibility of this study was ensured by participating others person who is expert in qualitative research. Peer debriefing was also done to assure the credibility of this research.

To ensure dependability of this research, an audit trial was made by giving vivid descriptions of the research process from the inception of the study to the report of study findings. The researcher providing thick, rich and structural textual descriptions ensured conformability. The researcher ensured transferability by providing evidence, detailed description of the study starting from sampling to data analysis.

### Data processing and analysis

Data was analyzed side by side on data collection. The analysis primarily focused on collected data in the form of expanded field notes and transcripts of recorded interviews. The digitally recorded data was listed carefully and transcribed to local language (Amharic) and conceptual translation was made. Images and sounds such as facial expressions, reluctance in responding to questions, emphatic nature of the responses, and frustrations in addressing certain issues were also systematically interpreted and their meanings noted on paper to be incorporated into the analysis. The transcribed data were read and re-read until full understanding of the meanings what participates said were gained to detect the emerged themes and sub-themes. After we read, primary coding was done. Primary coding consisted reviewing transcript line by line then secondary coding (axial coding) was performed to isolate key words and phrases to generate emerging sub-themes and themes. Displaying of data was done to identify the variation or richness of each themes. Data reduction was performed to get the overall collected data. Less broad themes were put together to form more broad themes and essential themes were separated from none essentials. Specifically, thematic analysis method was used to analyze data. Finally, the principal investigator interpreted the reduced themes and formed new themes to show the core meanings of the experiences. Quotes were used to highlight each category and show association with each theme. Atlas.ti qualitative software version 7 was used to facilitate data analysis.

### Ethical considerations

Ethical approval was obtained from Institutional Review Board (IRB) of College of Medicine and Health Science IRB protocol no 140/2021, Bahir Dar University. Informed verbal consent was obtained from each study participant and this verbal consent was kept in recorded for witness prior to the commencement of data collection. Helnscon declaration form was followed.

### Results

Sixteen study participants were participated in the study. Eight participants were male and eight were female. Their age ranges from 25–65 years. The mean age of the study participant was 41.7 years with ±10.7 SD. Eleven (68.75%) were married and the rest 31.25% were single at the time of interview. Fifteen (93.5%) study participants were from urban resident and one (6.24%) was from rural. Regarding to their religion, 15 (93.5%) were orthodox Christian and 1 (6.24%) was protestant.

(See Table 1 for detail sociodemographic variables of the study participants)

### Participants experience

The following main themes were explored with sub-themes: psychological experience, social experience, economical experience and anti-retroviral treatment interruption.

### Theme1: Psychological experiences

Most study participants shared experience of boredom, fear of being infected and Feeling of uncertainty about future ART drug availability. People whose family members were in foreign

**Table 1. Socio-demographic characteristics of the study participant (N = 16) who are on ART in Tach Gayint district in 2021.**

| Code | Sex | Age | Years with HIV | ART year | Educational status | Religion | Occupational status | Marital status | Resident |
|------|-----|-----|----------------|----------|--------------------|----------|---------------------|----------------|----------|
| P1 | F | 37 | 4 | 4 | Did not attend | Orthodox | Alcohol cashier | Divorced | Urban |
| P2 | M | 41 | 8 | 8 | Bachelor's Degree | Orthodox | Civil servant | Married | Urban |
| P3 | F | 25 | 3 | 2 | elementary | Orthodox | Housewife | Married | Urban |
| P4 | F | 65 | 10 | 10 | Did not attend | Protestant | no work | Widowed | Urban |
| P5 | F | 40 | 12 | 10 | Did not attend | Orthodox | no work | Divorced | Urban |
| P6 | F | 30 | 12 | 6 | Did not attend | Orthodox | Housewife | Married | Urban |
| P7 | F | 37 | 8 | 8 | Did not attend | Orthodox | Shopkeeper | Divorced | Urban |
| P8 | F | 30 | 4 | 4 | Secondary | Orthodox | Farmer | Married | Urban |
| P9 | M | 44 | 8 | 8 | Elementary | Orthodox | Farmer | Married | Rural |
| P10 | M | 30 | 2.5 | 2.5 | Diploma | Orthodox | Teacher | Married | Urban |
| P11 | F | 45 | 7 | 7 | Did not attend | Orthodox | Alcohol cashier | Divorced | Urban |
| P12 | M | 42 | 10 | 10 | Secondary | Orthodox | Barber | Married | Urban |
| P13 | M | 43 | 9 | 7 | Did not attend | Orthodox | Carpenter | Married | Urban |
| P14 | M | 55 | 5 | 5 | Secondary | Orthodox | Merchant | Married | Urban |
| P15 | M | 50 | 6 | 6 | Elementary | Orthodox | Farmer | Married | Urban |
| P16 | M | 40 | 11 | 8 | Secondary | Orthodox | Merchant | Married | Urban |

Note: M = MALE

F = female

Years = number years living with HIV since diagnosis

countries seemed to be more terrified and anxious. Most study participants reported COVID-19 brought negative psychological experience on them. They expressed feeling of anxiety, worried, depression, fear, tension, sadness and hopelessness. Three sub-themes namely fear of dying, feeling of uncertainty about future ART drug availability in the nearby health institutions and perceptions on severity of COVID-19 were identified.

## Fear of dying

Participants explained that COVID-19 made them to experience fear of dying in their future life. They heard the severity of COVID-19 is more in people with immune compromised patient and this made them to be in extreme fear. Due to fear of dying participants were exposed to psychological problem and they reported as they were not spending normal life due to the existence of the new virus called COVID-19.

"People said to me "HIV patients will die first if corona infects them . . ." when I hear this; I worry for myself and felt depressed. Internally I was demoralized at that time." A 43 years old male participant stated

Especially at the initial stage of the pandemic in our country, study participants mentioned that they were in extreme tension and lack of hopefulness in their future lives. They were assuming, as they will die immediately if the new virus contracted them.

n contrast to the above stated response of participants, few participants narrated that they did not feel anything related to the pandemic of COVID-19. Among of these one female participant explained, as she did not know the severity of COVID 19 on people living with HIV/AIDS.

### Feeling of uncertainty about future ART drug accessibility and availability

Participants narrated that they were uncertain what happened to their drug for the future. Most of them worried about the future availability of anti-retroviral therapy (ART) due to COVID-19. They mentioned shortage of drug for the future was their major concern. They stated, as they knew ART drug is imported from aboard and if exchanging of goods and services stopped with other countries, ART medication would not be easily available to them and this condition made them to be uncertain for their medication.

"I worry for the future about the availability of drug [ART]. If the drug is interrupted, we will not stay even for one day. This was my. . . [Stress]" 40 years female participant

### Perception on severity of COVID-19

Almost all participants explained as they accept the severity of COVID-19 on them. When they asked about severity of COVID19 on people living with HIV, most of the participants mentioned as it affect more in people having chronic disease including HIV.

Participants narrated that they would be severely affected by COVID-19 if they were infected. All most all interviewed participants except one knew the severity of the virus on them. They mentioned that people with low immunity including them were more risky than that of healthy individuals.

"Ehha . . . I do not tolerate the double burden of the two diseases. I will not cure if two and three virus infects me. When I meet with friends [PLWHA], I worried about the new virus as it kills us before HIV kills us but some people [PLWHV] said "No! No! It [corona] will not kill us". I said people without HIV can resist corona but I and other people who take drug [ART] will not survive." 40 years female participant

### Contexts (factors) related to psychological experience

**Personal contexts.** Knowledge on severity of disease and their immune status; Participants knew that their immunity status lead them for easily susceptibility for COVID-19. Occupational status of participant: merchants always move from place to place due to the nature of their work and this made them to fear of contracting and dying. Barber has many contacts with their customers and this cause to be fear of dying because they are easily at risk for contracting COVID-19.

**Familial contexts.** Low socioeconomic status of the households made them not to stay at home. Participants move here and there to win breads; this made them to be at risk for contracting COVID-19 and faced distressing psychological experiences.

**Contexts related to community and governments.** Lack of psychosocial and financial support, the information heard from the community about the myth and misconception about COVID-19 on people with living with HIV.

### Theme 2: Social experience

More than half of the study participant faced negative social relation in context of COVID-19 due to their HIV status. Some isolated themselves from social activities and others isolated the rest. COVID-19 made them to isolate themselves from participating in social activities like wedding ceremony and funeral ceremony because they worried about the contracting of COVID-19. Two sub themes emerged under this main theme:- Isolating themselves from social activities and Isolated by others

## Isolating themselves from social activities

Participants stated that they had isolated themselves from social activities especially during the initial stage of the pandemic in the country. Restricting themselves in social activities was explored due to fear of contracting the new virus called COVID-19.

> "As I tried to explain to you earlier, as soon as corona emerges and distributed throughout the country, we isolated ourselves from social life like absenting from funeral ceremony and people had not good attitude on me because I only know my health status and people do not know the reason why I absent from that social life. I was afraid to go to a place where people gather together" 30 years male participant

## Isolated by others

Few of the study participants reported as other people afraid and isolated them and them especially during the lock down period. Others assumed them as they are infected with COVID-19. As a result, other people become far away from them and the normal relation that was in pre-COVID-19 was reduced.

> "People afraid I because they have information as I am easily susceptible to the disease and assumed as I have already infected by the disease [COVID-19]. They afraid me and said be far away from us: living with hearing such type of sayings was difficult for me. So corona made my social life to be difficult." A 55 years old male

Amazingly in contrast to the above mentioned study participants, one female participant stated that as her social interaction with her neighbors increased. They assumed them themselves as they will die soon and stop fighting each other.

> "Yes before corona we have no good relation as there was (beletishign beletihugsh) type of thinking but after corona everything was changed. We said for this short period of time why we made sinful activities." A 37 female

## Contexts for social experiences

For the occurrence of these hostile social experiences, there were different reasons; these reasons classified as personal, familial and community; participants' knowledge on the severity of disease on them. Household contexts such as free moving of their family members in everywhere made them to face distressing experiences. Negative perception of community toward PLWHA, traditional activities of the community such as collecting in traditional activities of the community made some participants to face unsmooth relationships with people.

## Theme 3: Economic experience

Majority of the participant had varied experience of economic lose because of lockdown and isolation in COVID-19 pandemic. All participants reported that they were unable to get any financial support from the government and from the community in this COVID-19 pandemic period and their household income was reduced due to COVID-19. Under this, two sub-themes were identified 1) disruption of economic activity due to COVID-19 and 2) absence of financial support.

### Disruption of economic activity

Some participants brought their concern of significant loss in their business as a result of the closed market. In this study some people on ART reported as they stopped doing their work due to fear of the effect of COVID-19 on them. Two merchant participants express as they stopped shopping because they fear contracting of COVID-19. They mentioned their customers afraid them especially during the lockdown period and do not buy goods and service from them and this led to decrease their household income.

> "When corona virus emerges, market was closed at that time I unable to gain money because I stopped working and I stayed at home; in this case the household income decreased in some extent. For example, I was buying onion from farmers and selling it to urban residence but the market closed and I unable to gain any profit from the onion and I used the deposited money that was deposited for other purpose. As a result my household economy decreased." A 25 years old female

> "Most of the time I did not work because being hungry is somewhat better than death. I always feared corona because my activities as I told you made me at risk of corona. I stop cashiering and my income decreased." A 45 years female

### Absence of financial support

Participants stated that people living with HIV including them were economically disadvantaged group of population. However, they were not getting any financial support from anybody. The government, community and other social institution were not giving special financial support to people on ART in the study area.

> "Only god supports me. I am not getting any special support without God. In the last summer, I have one sack of wheat but this was given to all people [including those without HIV]. I did not gain any special support from anybody in this corona period. Few years back we were going to Debre Tabor and train different things and there was different support for us. Now a day everything is stopped. I have one daughter at home and she did not support me because she is not mature to support me and she did not remember corona. Only God supports me." A 65 years female

All participants were not gaining any financial support from the government and the community as a result they were living in extreme poverty and the emergency of COVID-19 pandemic made double burden on their financial problem.

### Contexts related to economic experience

Government context like lack of concern for people on ART in context of COVID-19 was their main reason for the above experience. Their occupation was the other factors for this distressing experience. Merchant participants experienced decrement household income but civil servants and farmers did not.

### Theme 4: ART drug interruption

Four participants explained that they had stopped collecting and taking ART drug though this was for a short period of time at the initial stage of the pandemic in our country (at the lockdown period). As they narrated, they were disturbed by the information about the disease

(COVID-19) and they considered health institutions were shifted giving routine service including ART to COVID-19 patients. They also feared and stopped going to health institution because they were thinking, as there were COVID-19 infected patients in the health center and going and making contact with them was the risk of contracting the virus. As a result, they stopped taking ART drugs though this was for short period.

> "I feared to come to this health center to take my drug as a result I interrupted drug intake for about one week because what I have brought is finished and when corona emerges I fear to go and brought. I assumed the health center was only giving for seriously ill patients and if I go to health center to brought drug, corona will infect me and I stopped coming to this health center." A 41 years old male participant

But study participants highlighted that due to negative effect of poor ART drug adherence, they were taking ART drug appropriately because ART is a lifesaving medication. They were collecting and taking their medication according to their appointment time even in the lockdown period. They preferred to take their ART medication and desired to continue healthy with HIV.

**Contexts.** Fear of contracting COVID-19 in the health facility, their personal perception about the health institution where they collected drug were factors for their drug interruption. Lack of full information about the routine service of the health center was one an additional reason for their drug intake interruption. Among the sixteen interviewed participants twelve study participants did not interrupted their ART drug and they were taking to increase their immune status. One female study participant reported as she was in intension to stop going and taking the drug during the lockdown period.

## Discussion

According to the findings, study participants had perceived the severity of COVID-19 on people with low immunity. As reports from studies in Indonesia and south Africa showed people living with HIV experienced feeling of hopelessness, stress and anxiety due to their HIV status alone [19, 24]. The current study revealed that COVID-19 created extra psychosocial problem in addition to HIV. So these findings indicated that people on ART confronted double psychosocial problems because people living with HIV/AIDS and taking ART has other concern in addition to COVID-19 when compared to normal population [25]. Their extreme worried and tension was due to fear of contracting COVID-19 because of the severity of disease on chronic patients. Physical distancing or social isolation recommended by center of disease control (CDC) to lower the spread of COVID-19 may add additional burden to already highly burdened lives. This finding is also supported by a qualitative study done in Ethiopia on adolescent people living with HIV that narrated as they stopped attending medias due to fear of information transmitted about the huge burden of COVID-19 [23]. Fear of dying due to COVID-19 infection and uncertainty about future ART drug availability and accessibility was the major concern of this research finding. This finding is in lined with following two researches done in Belgium and Brazil [21, 26]. Some study participants in this study and study participants mentioned from the above heard about the severity of COVID-19. This was the reason why they faced psychosocial problems. In contrast to this few participants in this study do not feel any negative psychological problem related to COVID-19. When they asked the reason why not they felt anything they explained the reason and this reason highlight as they do not have deep information about COVID-19. The other reason why not they fell any psychosocial problem was they have been already hopelessness as a result of HIV infection.

Study participants in this research worried about future ART drug accessibility and availability. This finding is similar with the finding of the study done in Europe (central and eastern), Uganda and Ghana on people living with HIV [15, 27, 28]. Few participants in the study reported that they feared to go to health institute and stopped taking ART drug for short period of time during the lock down period. This finding is nearly similar with study done in other study which showed poor HIV clinic attendance, poor ART adherence [29]. ART drug intake and consistent adherence has improved the long-term health outcomes among PLWH, but they are vulnerable to severe health problems if interruptions in treatment occur due to the COVID-19 pandemic. If people living with HIV did not take ART their viral load is more likely to increase [30], leading to lower CD4-count and an increase the risk of developing opportunistic infection [31]. As such, it is imperative people living with HIV infection need to engaged and health care providers amidst COVID -19 pandemic to insure consistent in HIV related care and treatment. This ART drug interruption due to COVID-19 pandemic may be absence of different support by the government and community to PLWHA. In this COVID-19 pandemic period, people infected with HIV need special psychosocial support because they have other health problem (HIV) which pushed them to different psychosocial problems like tension, anxiety, depression and lose of hopefulness.

Study participants in this study explained the disease and its consequences exacerbated their economic crisis. This finding is in lined with the study done in Argentina [32]. This showed that people ON ART are living in low socio-economic status and the economic burden of COVID-19 severe in these groups of population because these populations are disproportionally affected by COVID-19. They fear feared contracting the new virus, restricted their economic activities, and stayed at home especially during the lockdown period made an additional factor for their decrement of household income. However, COVID-19 did not reported to bring considerable change in their economic changes on some people on ART especially who are rural dwellers and civil servants in this research. The reason for this is that people on ART did not restrict in different income generating activities. According to the study findings in western Kenya and Zimbabwe losing employment, not being able to get access their job, unable to pay money for food and slowdown of any business activities were the common problems among people on ART [22, 33]. Some of these problems like difficulty of paying money for goods and service were highlighted among people on ART in COVID-19 pandemic and this research area and the finding was in lined with the result done in the above researches. This similarity may be due to the fact that COVID-19 brought difficulty of exchanging goods between countries and the production of these goods were decreased. The other reason for this is that people living with HIV are not economically supported by the government, other non-government organizations including the community. However, losing of employment and unable to get job opportunity due to COVID-19 were not the problem in this research. This discrepancy occurred due to the fact that study participants in this research were not absolutely restricted in their income generating activities and study participants in this research were already had engaged in their business activities and were not waiting job opportunities. The government was giving free health and health related service to those people living with HIV in the study area but this was contradict with the other research done that showed insured people are getting easily access of care than that of non-insured people [34]. In the research area people living with HIV are getting free health and health related service and this indicate that during COVID-19 pandemic people on ART are not face hardship of getting health access if they seek the service.

Some participants mentioned as they restrict themselves from visiting ART clinic and stopped taking ART. The reason was their fear of contracting the virus in the health institution. They may assumed the health institutions were shifting their routine service to COVID-19 and

the information disseminated about the severity of disease made them to be stressed and stopped visiting ART outpatient department. The result of this study was similar with the study done in Uganda which showed participants feared to go to health institutions to collect their ART drugs [28]. A qualitative study that was done in Ethiopia on adolescent living with HIV revealed that these adolescent interrupted their intake drug due to their assumptions as there was no routine ART service in the health facilities as a result they stopped going to health institution and interrupted their drug intake. A study done in China which showed people wanted to conceal their status from their families [14, 23] supplemented the findings of this research. However, the reasons for this similarity were different with the study in China. In China, study participants wanted to conceal their status from their families because the government faced challenging condition to supply ART medication service face to face and used mailing of ART services. However, in this study, the causes for interruption were participants' fear of contacting someone who was infected with COVID-19 and their assumptions as there were no routine services including ART in the health facilities.

## Limitation of the study

This study was conducted after the lockdown period; it would be better if it was conducted at time of lockdown for better understanding their experiences. This led participants to recall bias. The study couldn't access enough participants from rural areas and this led difficulty of exploring their experience in rural contexts. Some people on ART did not agree to be interviewed. They might have high level of experiences if they were explored. The limitation of this study was small sample size and participants were taken only from one health center and taking these findings hinder generalizability of the result. This study used phenomenological study design and establishing validity, reliability was challenging, and subjectivity may not be completely avoided.

Further research on myth and misconception of people with HIV/AIDS and COVID-19 should be considered.

## Conclusion

In this descriptive phenomenological study, the lived experiences of people on ART in context of COVID-19 pandemic were explored. Main themes and sub-themes were emerged from the interview. The main themes were psychological experience, change in social interaction, economic hardship and interruption of ART drugs intake. Hostile personal, household and community contexts leading the above distressing experiences were also explored. HIV only by itself creates psychosocial problems and people on ART were not gaining support of it in COVID-19 pandemic period. They also faced economic problem as a result few of them were living in challenging condition to fulfill the household food security. The social interaction of people on ART was compromised as they isolate themselves from different social activities and people suspect them as they are already infected with COVID-19. In the lock down period few participants explained as they interrupted ART drug intake. The contexts for their drug interruption were that they feared the risk of contracting the virus at the health facility and they considered health institution had shifted from their routine activity to caring of COVID-19 patients. As they narrated any type of support was not given to them. All participants stated as they need different type of support because they were disadvantaged group of the population.

## Recommendations

People on ART are living with many psychological problems and they deserve psychological support. The double burden of the two viruses made them to be in psychological problem. So

it is recommended to local government to create favorable environment for them to reduce hostile contexts which made distressing psychological experiences.

People on ART were living extreme poverty and they deserve financial support. COVID-19 added other financial crisis on them. The government and local community including traditional institutions are advised to give priority for people on ART when different aids were given to people. Giving extra quota of aids, making conducive environment for people on ART to engage in income generating activities is very substantial.

To reduce unfavorable psychosocial and economic experiences of COVID-19 on them, linking different governmental and non-governmental organizations to work on different concern of people on ART is very important.

Health professionals are advised to give accurate information to people on ART and the community about the relationship between COVID-19 and HIV to voiding unnecessary perception of COVID-19.

Further research on the myth and misconception of community about the relationships between people with HIV and COVID-19 is very important.

## Supporting information

**S1 File. Codes and quotations with primary documents.**
(RTF)

## Acknowledgments

We would like to say thank you our study participants, Tach Gayint woreda health office and Amhara regional health bearuea. We would also like to say thank you Tach Gayint woreda communication office who help as in recording of the interview.

## Author Contributions

**Conceptualization:** Tadele Derbew Kassie, Yosef Wasihun, Abiot Aschale, Fentie Ambaw.

**Data curation:** Tadele Derbew Kassie, Yosef Wasihun, Abiot Aschale, Fentie Ambaw.

**Formal analysis:** Tadele Derbew Kassie, Yosef Wasihun, Abiot Aschale, Fentie Ambaw.

**Funding acquisition:** Tadele Derbew Kassie, Abiot Aschale, Fentie Ambaw.

**Investigation:** Tadele Derbew Kassie, Abiot Aschale, Fentie Ambaw.

**Methodology:** Tadele Derbew Kassie, Fentie Ambaw.

**Project administration:** Tadele Derbew Kassie.

**Resources:** Tadele Derbew Kassie.

**Software:** Tadele Derbew Kassie, Yosef Wasihun, Fentie Ambaw.

**Supervision:** Tadele Derbew Kassie, Yosef Wasihun, Fentie Ambaw.

**Validation:** Tadele Derbew Kassie, Yosef Wasihun, Abiot Aschale, Fentie Ambaw.

**Visualization:** Tadele Derbew Kassie, Yosef Wasihun, Abiot Aschale, Fentie Ambaw.

**Writing – original draft:** Tadele Derbew Kassie, Yosef Wasihun, Abiot Aschale, Fentie Ambaw.

**Writing – review & editing:** Tadele Derbew Kassie, Yosef Wasihun, Abiot Aschale, Fentie Ambaw.

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
