## [Decision Letter · Decision Letter 0]

9 Dec 2022

PONE-D-22-25059Lived experience of people on anti-retro viral therapy in the context of covid-19: a phenomenological study

PLOS ONE

Dear Dr. kassie,

Thank you for submitting your manuscript to PLOS ONE. After careful consideration, we feel that it has merit but does not fully meet PLOS ONE’s publication criteria as it currently stands. Therefore, we invite you to submit a revised version of the manuscript that addresses the points raised during the review process.

Please note that we have only been able to secure a single reviewer to assess your manuscript. We are issuing a decision on your manuscript at this point to prevent further delays in the evaluation of your manuscript. Please be aware that the editor who handles your revised manuscript might find it necessary to invite additional reviewers to assess this work once the revised manuscript is submitted. However, we will aim to proceed on the basis of this single review if possible. 

We look forward to receiving your revised manuscript.

Kind regards,

Alice Coles-Aldridge

Editorial Office

PLOS ONE

Journal Requirements:

2. Thank you for submitting the above manuscript to PLOS ONE. During our internal evaluation of the manuscript, we found significant text overlap between your submission and previous work in the [introduction, conclusion, etc.].

Please revise the manuscript to rephrase the duplicated text, cite your sources, and provide details as to how the current manuscript advances on previous work. Please note that further consideration is dependent on the submission of a manuscript that addresses these concerns about the overlap in text with published work.

[If the overlap is with the authors’ own works: Moreover, upon submission, authors must confirm that the manuscript, or any related manuscript, is not currently under consideration or accepted elsewhere. If related work has been submitted to PLOS ONE or elsewhere, authors must include a copy with the submitted article. Reviewers will be asked to comment on the overlap between related submissions (http://journals.plos.org/plosone/s/submission-guidelines#loc-related-manuscripts).]

We will carefully review your manuscript upon resubmission and further consideration of the manuscript is dependent on the text overlap being addressed in full. Please ensure that your revision is thorough as failure to address the concerns to our satisfaction may result in your submission not being considered further

3. "PLOS requires an ORCID iD for the corresponding author in Editorial Manager on papers submitted after December 6th, 2016. Please ensure that you have an ORCID iD and that it is validated in Editorial Manager. To do this, go to ‘Update my Information’ (in the upper left-hand corner of the main menu), and click on the Fetch/Validate link next to the ORCID field. This will take you to the ORCID site and allow you to create a new iD or authenticate a pre-existing iD in Editorial Manager. Please see the following video for instructions on linking an ORCID iD to your Editorial Manager account: " ext-link-type="uri" xlink:type="simple">https://www.youtube.com/watch?v=_xcclfuvtxQ"

4. "In your Data Availability statement, you have not specified where the minimal data set underlying the results described in your manuscript can be found. PLOS defines a study's minimal data set as the underlying data used to reach the conclusions drawn in the manuscript and any additional data required to replicate the reported study findings in their entirety. All PLOS journals require that the minimal data set be made fully available. For more information about our data policy, please see http://journals.plos.org/plosone/s/data-availability.

We will update your Data Availability statement to reflect the information you provide in your cover letter."

Additional Editor Comments (if provided):

Reviewers' comments:

Reviewer's Responses to Questions

**Comments to the Author**

1. Is the manuscript technically sound, and do the data support the conclusions?

Reviewer #1: Yes

2. Has the statistical analysis been performed appropriately and rigorously? 

Reviewer #1: Yes

3. Have the authors made all data underlying the findings in their manuscript fully available?

Reviewer #1: Yes

4. Is the manuscript presented in an intelligible fashion and written in standard English?

Reviewer #1: No

5. Review Comments to the Author

Reviewer #1: This is a good paper. It is novel, especially with the unique perspective it brings in the light of COVID-19 impacts on the experience of patients. However, there are many grammatical errors strewn around a large swathe of the paper.

There are additional gaps in the paper. The purpose of the study was not well-expressed under the introduction.

The authors did not also summarize the demographics of the participants appropriately. For example, how many of the participants were married and single at the time of the interview? This was not provided. What is the mean age of the candidates sampled? How many of them were living in urban area compared with those in rural areas. You will have to start counting the data on Table 1 to be able to do this.

There are other issues with labeling and ambiguity with usage of words. Participant 12 is labeled as Me instead of M. Educational status is tagged as degree. What kind of degree? Bachelor's, Master's or a PhD, or are they lumped together? The author stated that all study participants with age 18 years or more were selected purposively but elsewhere they mentioned 25-65. I also wonder why they selected the age sample of 25-65. Across many countries, and other studies, there is a considerable population on ART who are below 25. Targeting them with interventions and studies like this would help us with additional contributions to knowledge which helps us to achieve epidemic control. Interviewing this cohort would have enriched this paper and possibly offer additional perspective.

The authors did not compare their results properly with was done elsewhere. For example, while they referenced studies done in Uganda, Ethiopia and China, the comparison is not specific and detailed enough to know what was done in those countries, and how their interventions and findings differ from what the authors had.

Under the limitation, the authors did account for what they do with "shy people" who could not completely share their experiences. Secondly the authors said rural residents were not accessed. One of the participants coded P9, was accessed and from the rural area. The author could have written that they couldn't access enough participants from rural areas rather than saying that they were not accessed, or is that the status 'rural' for that client was an error or an interpolation?

The authors did not make any recommendations based on their findings.

If the authors can make corrections to the gaps highlighted, and do extensive review to redress the grammatical errors in the paper, the study would be a good one.

6. PLOS authors have the option to publish the peer review history of their article (what does this mean?). If published, this will include your full peer review and any attached files.

Reviewer #1: **Yes: **Peter Okpe Agada

---

## [Author Response · Author response to Decision Letter 0]

20 Jan 2023

We express our cordial gratitude for the valuable constructive reviews of our paper entitled with “the lived experience of people on ART in context of COVID-19 in Tach Gayint district a phenomenological study”. The reviewer’s comments have helped us to further strengthen the overall quality of the paper. With great care, we have incorporated all the suggestions/corrections as proposed by the reviewer. The specific responses to the reviewer’s comment are listed below.

Comment 1: The purpose of the study was not well-expressed under the introduction. 

Response: The manuscript addressed this issue and updated.

Comment 2: The authors did not summarize the demographics of the participants appropriately. 

 Response: it is very important concept and we saw it and summarized.

Comment 3: with labeling and ambiguity with usage of words. Participant 12 is labeled as Me instead of M. Educational status is tagged as degree. What kind of degree? Bachelors’, Master’s or a PhD, or are they lumped together? 

Response; it is very constructive comment and we updated it in the paper 

Comment 4: The authors did not compare their results properly with was done elsewhere.

Response: we saw it detail and it is updated.

Comment 5: Under the limitation, the authors did account for what they do with "shy people" who could not completely share their experiences. Secondly the authors said rural residents were not accessed. One of the participants coded P9, was accessed and from the rural area. The author could have written that they couldn't access enough participants from rural areas rather than saying that they were not accessed, or is that the status 'rural' for that client was an error or an interpolation?

Response we admired giving such type of constructive comment and we incorporated the issue given.

 Comment 6: The authors did not make any recommendations based on their findings.

Response: we revised it and tried to match the result with recommendation 

Sincerely

---

## [Decision Letter · Decision Letter 1]

12 Mar 2023

PONE-D-22-25059R1Lived experience of people on anti-retro viral therapy in the context of covid-19: a phenomenological studyPLOS ONE

Dear Dr. kassie,

Thank you for submitting your manuscript to PLOS ONE. After careful consideration, we feel that it has merit but does not fully meet PLOS ONE’s publication criteria as it currently stands. Therefore, we invite you to submit a revised version of the manuscript that addresses the points raised during the review process.

If applicable, we recommend that you deposit your laboratory protocols in protocols.io to enhance the reproducibility of your results. Protocols.io assigns your protocol its own identifier (DOI) so that it can be cited independently in the future. For instructions see: https://journals.plos.org/plosone/s/submission-guidelines#loc-laboratory-protocols. Additionally, PLOS ONE offers an option for publishing peer-reviewed Lab Protocol articles, which describe protocols hosted on protocols.io. Read more information on sharing protocols at https://plos.org/protocols?utm_medium=editorial-emailutm_source=authorlettersutm_campaign=protocols.

We look forward to receiving your revised manuscript.

Kind regards,

Boshra Ismael Ahmed Arnout

Academic Editor

PLOS ONE

Additional Editor Comments:

Dear Author

The paper PONE-D-22-25059R1 has been reviewed by experts in the field who consider that the revised paper can publish after major revision. For your guidance, you can benefit from the reviewer's comments are appended below.

We wish you a meaningful day.

Yours Sincerely

Reviewers' comments:

Reviewer's Responses to Questions

**Comments to the Author**

1. If the authors have adequately addressed your comments raised in a previous round of review and you feel that this manuscript is now acceptable for publication, you may indicate that here to bypass the “Comments to the Author” section, enter your conflict of interest statement in the “Confidential to Editor” section, and submit your "Accept" recommendation.

Reviewer #2: (No Response)

2. Is the manuscript technically sound, and do the data support the conclusions?

Reviewer #2: Partly

3. Has the statistical analysis been performed appropriately and rigorously? 

Reviewer #2: No

4. Have the authors made all data underlying the findings in their manuscript fully available?

Reviewer #2: Yes

5. Is the manuscript presented in an intelligible fashion and written in standard English?

Reviewer #2: Yes

6. Review Comments to the Author

Reviewer #2: After reading the article, I thank the researchers for such a good job, and add some remarks here

1- The researchers did not provide good detailed information about the study design

- It is important to describe the participants in a separate part, omitting the sample size part.

- Deletion of the name (method and procedures of data collection) and amending it to (the context of the study).

- There is an overlap between the elements of the paper (a note that needs great attention)

The procedures of credibility and reliability are not clear, presented in a narration without linking to the study

The steps of qualitative analysis presented superficially are not sufficient

Presentation of results requires focus

- Add study limitations and future studies

7. PLOS authors have the option to publish the peer review history of their article (what does this mean?). If published, this will include your full peer review and any attached files.

Reviewer #2: No

---

## [Author Response · Author response to Decision Letter 1]

27 Mar 2023

plosone is very friendly to use and editors are humble. so I always prefer your journal to send my work for publication.

---

## [Decision Letter · Decision Letter 2]

15 May 2023

Lived experience of people on anti-retro viral therapy in the context of covid-19: a phenomenological study

PONE-D-22-25059R2

Dear Dr. kassie,

We’re pleased to inform you that your manuscript has been judged scientifically suitable for publication and will be formally accepted for publication once it meets all outstanding technical requirements.

Kind regards,

Boshra Ismael Ahmed Arnout

Academic Editor

PLOS ONE

Additional Editor Comments (optional):

Reviewers' comments:

Reviewer's Responses to Questions

**Comments to the Author**

1. If the authors have adequately addressed your comments raised in a previous round of review and you feel that this manuscript is now acceptable for publication, you may indicate that here to bypass the “Comments to the Author” section, enter your conflict of interest statement in the “Confidential to Editor” section, and submit your "Accept" recommendation.

Reviewer #2: (No Response)

2. Is the manuscript technically sound, and do the data support the conclusions?

Reviewer #2: Yes

3. Has the statistical analysis been performed appropriately and rigorously? 

Reviewer #2: Yes

4. Have the authors made all data underlying the findings in their manuscript fully available?

Reviewer #2: Yes

5. Is the manuscript presented in an intelligible fashion and written in standard English?

Reviewer #2: Yes

6. Review Comments to the Author

Reviewer #2: (No Response)

7. PLOS authors have the option to publish the peer review history of their article (what does this mean?). If published, this will include your full peer review and any attached files.

Reviewer #2: No

---

## [Editor Report · Acceptance letter]

2 Jun 2023

PONE-D-22-25059R2 

Lived experience of people on anti-retro viral therapy in the context of covid-19: a phenomenological study 

Dear Dr. Kassie:

I'm pleased to inform you that your manuscript has been deemed suitable for publication in PLOS ONE. Congratulations! Your manuscript is now with our production department. 

Kind regards, 

on behalf of

Professor Boshra Ismael Ahmed Arnout 

Academic Editor

PLOS ONE